

**A comparative study of auroral morphology distribution between northern and**
**southern hemispheres based on automatic classification**
Qiuju Yang[1], Ze-Jun Hu*[2]
[1]School of Physics and Information Technology, Shaanxi Normal University, Xi'an, 710119, China;
[2]SOA Key Laboratory for Polar Science, Polar Research Institute of China, Shanghai, 200136, China;
*Correspondence to*: Ze-Jun Hu (huzejun@pric.org.cn)
**Abstract:** Aurora is a very important geophysical phenomenon in the high latitude of Arctic and
Antarctic regions, and it is significant to make a comparative study of the auroral morphology between
the two hemispheres. Based on the morphological characteristics of the four labeled dayside auroral
types (include auroral arc, drapery corona, radial corona and hot-spot aurora) on the 8001 dayside
auroral images at Chinese Yellow River Station in 2003, and by extracting the local binary pattern
(LBP) features and using k-nearest classifier, this paper makes an automatic classification to the 65361
auroral images of the Chinese Yellow River Station during 2004-2009 and the 39335 auroral images of
the South Pole Station between 2003-2005, and finally obtains the occurrence distribution of the
dayside auroral morphology in northern and southern hemispheres. The statistical results indicate that
the four auroral types present similar occurrence distribution between the two stations. To the best of
our knowledge, we are the first to report the statistical comparative results of dayside auroral
morphology distribution between northern and southern hemispheres.
**Keywords:** auroral images, morphology distribution, automatic classification
**1 Introduction**
Aurora is caused by the collision of solar wind charged particles with the atoms in the polar
ionosphere. Study on the morphology and evolution of aurora is not only help to reveal the solar
wind-magnetosphere-ionosphere coupling processes and their internal mechanism, but also provides
important physical principles for the space weather forecast (Nishimura et al., 2010; Hu et al., 2009,
2010, 2012, 2013, 2017a, 2017b; Han et al., 2015, 2016, 2017).
As auroral research continues, auroral comprehensive observation has become an important polar
scientific research program for various countries in the world. The all-sky Camera (ASC), which has
high spatial resolution and can make broad view and long-time continuous observation, is one of the
most effective tools for auroral observation and the inversion of magnetic layer structure and dynamic
process. Many countries have established their ground-based observation systems, such as Norway,
Sweden, America, and Canada in Arctic regions, while Japan, England, America, and Australia in
Antarctic regions. In recent years, our country has made great progress in the field of auroral
observation: not only have the Zhongshan station and Yellow River station been established in
Antarctic and Arctic respectively, but the joint aurora observatories have been respectively established
with Norway and Iceland as well.
With the establishment of many ground-based stations, those optical instruments produce





hundreds of millions of images annually (Syrjäsuo and Partamies, 2012), causing automatic auroral
image analysis techniques urgently needed (Syrjäsuo et al., 2001; 2004; 2007). Up to now, there are
many studies conducted on one single station: Syrjäsuo and Donovan (2004) first explored computer
vision techniques to auroral image classification, they analyzed 350,000 Gillam ASC images of the
CANOPUS (Canadian Auroral Network for the OPEN Program Unified Study) program during
1993-1998, and categorized them into no aurora, arcs, patchy aurora, omega-band, and other auroral
structures. Based on the ASC data observed in Kilpisjärvi, Syrjäsuo and Partamies (2012) made an
automatic detection of aurora (aurora exist or not?). Ebihara et al. (2007) analyzed the quasi-stationary
auroral patches based on the ASC images observed between ~0900 and ~1400MLT at the South Pole
Station. In recent years, ASC images in Chinese Yellow River Station have been widely studied. Based
on the characteristics of the auroral spectra and morphology, Hu et al. (2009) partitioned the dayside
oval into four auroral active regions and further classified the dayside aurora into arc, drapery corona,
radial corona and "hot-spots". Wang et al. (2010) and Yang et al. (2012) have proved that the local
binary pattern (LBP) can well represent the complex morphology of aurora, and the
multiple-wavelength intensity distributions are further confirmed by automatic classification.
However, each of the aforementioned studies was performed on one station; although MIRACLE
network (Syrjäsuo et al., 1998) includes several stations, the data were studied as a whole (Rao et al.,
2014; Savolainen et al., 2016). Recently, Pulkkinen et al (2011) reported the auroral occurrence by
using auroral observations from 5 stations in Fennoscandia and Svalbard in 2000~2009; and Partamies
et al (2014) used 1 million auroral images captured at five camera stations in Finnish and Swedish
Lapland in 1996~2007 to study the solar cycle and diurnal dependence of auroral structures, the data in
different stations were still analyzed together. Based on the synoptic distribution of the average auroral
intensity in the Arctic and Antarctic, Hu et al. (2014) discussed the hemispheric asymmetry of the
dayside auroral oval structures, yet they did not consider auroral morphology limited to manual
analysis. At present, there are very few comparative studies based on multi stations, especially the
contrastive study about auroral morphology between northern and southern hemispheres. **The different**
**dynamic processes in the magnetosphere can result different morphological characteristics of the**
**aurora. Through comparing the morphological characteristics of the auroras between the**
**northern and southern hemispheres, investigator can study the difference or similar of**
**ionospheric responses between the northern and southern hemispheres, which result from the**
**dynamic processes in the magnetosphere.** In this paper, the LBP descriptor is exploited to
characterize the ASC images, and the $k$-nearest neighbor ($k$-NN) classifier (Theodoridis and
Koutroumbas, 2006) is used to make a statistical comparative analysis of the dayside auroral
morphology distribution, with the data captured by the ASCs located at the Antarctic South Pole
Station (SPS) and the Arctic Yellow River Station (YRS).
The remainder of this paper is organized as follows. In Section 2, the data and LBP-based
representation method are introduced. The automatic recognition experimental results of dayside aurora
in SPS and YRS are presented in Section 3. Section 4 is the discussion. Finally, the conclusion is drawn
in Section 5.



## 2 Data and Methodology

### 2.1 Data introduction and pre-processing

The auroral data explored in this paper were captured by the all-sky cameras at two stations. (1) The Japanese South Pole Station (SPS) at South Pole: SPS is located at 90.0 °S geographic latitude and −74.3 ° corrected geomagnetic latitude in Antarctica (Ebihara et al., 2007), where MLT~UT-3.6 hrs. (2) The Chinese Yellow River Station (YRS) at Ny-Ålesund, Svalbard (Hu et al., 2009): YRS is located at geographic coordinates 78.92 °N, 11.93 °E and corrected geomagnetic latitude 76.24 °, where MLT~UT-3.6 hrs.

### 2.1.1 South Pole Station Data

The optical instruments at SPS can make 24-h surveys of auroral emissions with a temporal resolution of a few seconds to dozens of seconds during the winter season from April to August. In this paper, we focus on the dayside aurora (0900–2200 UT/0524–1824 MLT) from May to August to avoid daylight influence. Optical instruments cannot work normally in cloudy or foggy weather, and after eliminating those invalid data captured under bad weather conditions, altogether 211-day auroral observations are selected from May 2003 to August 2005 to constitute the SPS data set, named as auroral test dataset 1 (ATD1), which consists of 39335 images.

Prior to analyzing these ASC images, some preprocessing steps are performed: (1) Rescaling. There are few pixels' intensity greater than 8000, therefore all images are stretched with a cutoff value of 8000. Image stretching can preserve pixels' relative intensity and enhance image contrast, making auroral images easier to be classified. In addition, the file format of auroral images is converted from TIF to PNG for the convenience of processing with computer. (2) Cropping and rotation. We first crop the images to make the central field of view be the center of the ASC images; and considering the majority of auroral structures are east-west direction (Kauristie et al., 2001), the images are rotated counter-clockwise by 125.57 ° to make the south direction upward; then, given that the edges of ASC images have reached the maximum deformation rate of panoramic cameras, a circle mask with a radius of 199 pixels is applied to cut off the circumferential edge regions where wide-angle distortion are serious and may contain SPS lights. After that, the ASC images are cropped from 512×512 to 398×398 pixels. (3) Radial turnover in the direction of east and west, in order to make images keep consistent with that of YRS (left-east, right-west).

### 2.1.2 Yellow River Station Data

The ASCs at YRS continuous to produce auroral images on three wavelengths (427.8, 557.7 and 630.0 nm) for 24h per day with a temporal resolution of 10 s during the whole winter season (October to March). In this paper, we concentrate on the dayside aurora (0300–1500 UT/0600–1800 MLT) at 557.7 nm from November to February. After removing those images captured under bad weather conditions, altogether 249-day auroral data are selected from December 2003 to January 2009 to constitute the YRS data set. It is divided into two parts: 1) Auroral train dataset (ARD), consists of 8001 images captured from December 2003 to January 2004. 2) Auroral test dataset2 (ATD2), includes 65361 images generated from October 2004 to February 2009.

Also some preprocessing steps are conducted to YRS images. (1) Removing system noise. System noise in ASC instruments are caused by dark current, the value of which is 564 for the wavelength of

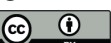


557.7nm auroras. (2) Masking and cropping. In order to have the same size with the ASC images at
SPS, a circle mask with a radius of 199 pixels is used to cut off the outer regions and the image size is
cropped to 398×398 pixels. (3) Rotation. The images are rotated counter-clockwise by 61.1 ° to make
north direction upward, which helps the division of images using vertical and horizontal lines at the
step of image representation. (4) Rescaling. After analysis, all images are stretched with a cutoff value
of 4000. Note although the Rayleigh intensity is different between YRS images and SPS images, it
does not influence the following results since we extract the texture feature of ASC images as follows.
**2.2 ASC image representation**
The original LBP operator was introduced by Ojala et al. (1996) primarily for texture
classification, and we have proved it has powerful ability to characterize the spatial texture of auroral
images (Wang et al., 2010; Yang et al., 2012). In this paper, the LBP operators with a partition scheme
are applied to represent ASC images.
LBP is a simple and efficient texture descriptor; it characterizes a local region by comparing the
relative gray values between the central pixel and its neighboring pixels. Figure 1 shows the calculation
process of the basic LBP operator. The central pixel value is 100, and its 3×3 neighborhood pixels are
80,90,120,110,130,120,100,70; if the gray value of the neighboring pixel is higher than the central
pixel, the threshold value is set to 1, otherwise to 0; therefore the results are 0,0,1,1,1,1,1,0. The
sequence is treated as a binary number, and by assigning the weight $2^i$ for the $i$th pixel in the
neighborhood, the binary sequence is converted to a decimal number 62. Similar processing is
performed to all the pixels in the image, and the decimal results are made histogram statistics.
According to the characteristics of auroral images, each image is divided into 3 rows and 6 columns
and obtain 18 rectangular blocks (Figure 2 (a)), and the histogram of the LBP patterns is calculated in
each block (Figure 2 (b)). Finally, these individual histograms are concatenated into a global descriptor
for ASC images (Figure 2 (b)).

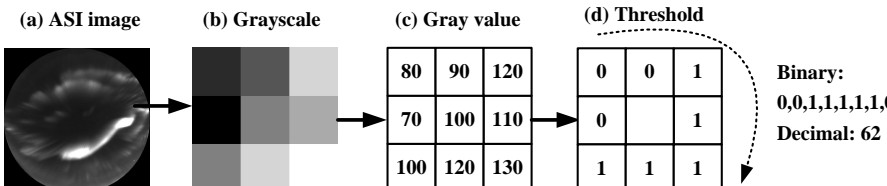

**Figure 1. Calculation process of the basic LBP operator.**



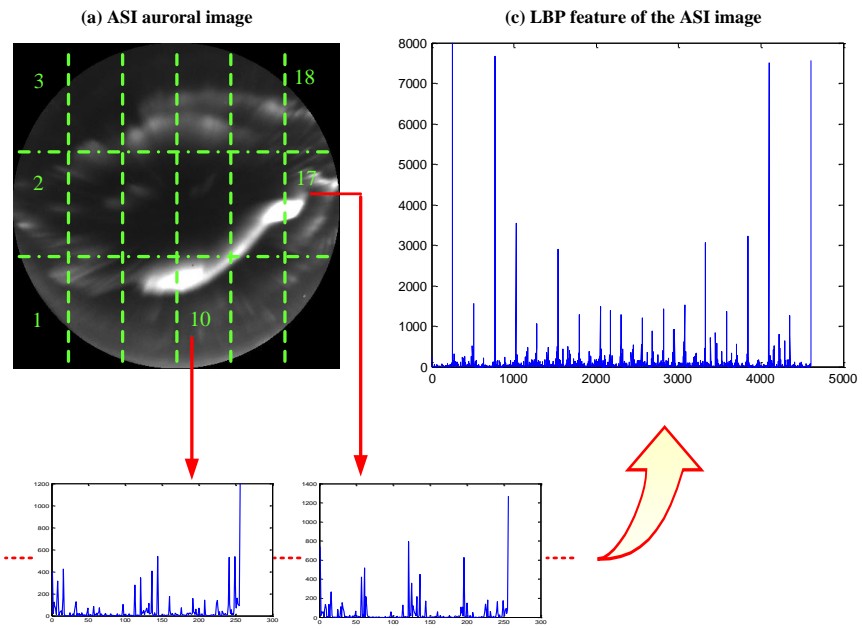

**(b) LBP histogram of each block**

**Figure 2. The LBP extraction process of the auroral image. The image showed was captured at 05:31:51 on December 21, 2003.**

**3 Automatic recognition of dayside aurora at SPS and YRS**

**3.1 Classification mechanism and data labeling**

In this paper, the auroral images are classified into arc, drapery corona, radial corona, and "hot-spot" based on the spectral and morphological signatures (Hu et al., 2009). (1) Arc aurora is a striped auroral form with east-west extended and narrow north-south spanning characteristics, and often multiple auroral arcs appear simultaneously in the sky. (2) Drapery corona is a weak display with the features of east-west elongated bands, and often multiple parallel rays appear at the same time. (3) Radial corona has clear radial-like structures which spread from the zenith in all directions. (4) "Hot-spot" aurora has complex structures, showing rayed-like coronal auroras, irregular patches of auroral intensity enhancement (vortex, spots) and arc-like auroral mixed morphologies. **In previous investigations (Hu et al., 2009; Wang et al., 2010), the investigators found that these auroras mainly appear in the "hot-spot" auroral active region, so they are named as "Hot-spot" aurora.** More detailed descriptions about the four auroral types can be referred to (Hu et al., 2009; Wang et al., 2010).

Based on the above descriptions, and referring to the synchronous ASC images at wavelengths of 427.8 and 630.0 nm (Wang et al., 2010), ASC images in ARD dataset are manually labeled as the abovementioned four categories. In order to avoid the very similar morphology between adjacent ASC images (because of the short sampling interval of 10s at YRS), the ARD dataset are constructed by extending the interval between adjacent images to ~1minute.



165

## 3.2 Auroral morphology recognition at SPS and YRS

### 3.2.1 Image Retrieval

The content-based image retrieval experiments are performed on ARD and ATD1 to examine the morphology difference between auroral images at YRS and SPS. The retrieval results are visually estimated whether each retrieved image has similar auroral morphology with its query image. Chi-square ($\chi^2$) histogram distance is used as a matching criterion. The smaller the distance, the more similar are the two images. The distance is defined as

$$\chi^2(p,q) = \sum_i (p_i - q_i)^2 / (p_i + q_i) ,\qquad(1)$$

where $p$ is the LBP histogram of the query image, $q$ is that of the retrieved image, and $i$ indicates the index of feature vector ($i$=1~4608). For a given query image in ARD, the matching image in ATD2 is the one with the smallest $\chi^2$ distance to the query image. Figure 3 shows the query set, whose images are sampled from ARD of YRS, and by searching the whole ATD1 dataset, the most similar retrieved result is exported. For each of the four auroral types, Figure 3 shows three image pairs. From Figure 3, we can conclude that each query and retrieved image pair is of the same auroral type with similar but not identical auroral morphology. The fact that each image pair belongs to the same category indicates the effectiveness of our approach, and the morphology differences are resulted from the observation difference itself between YRS and SPS and the error caused by the insufficient algorithm accuracy.





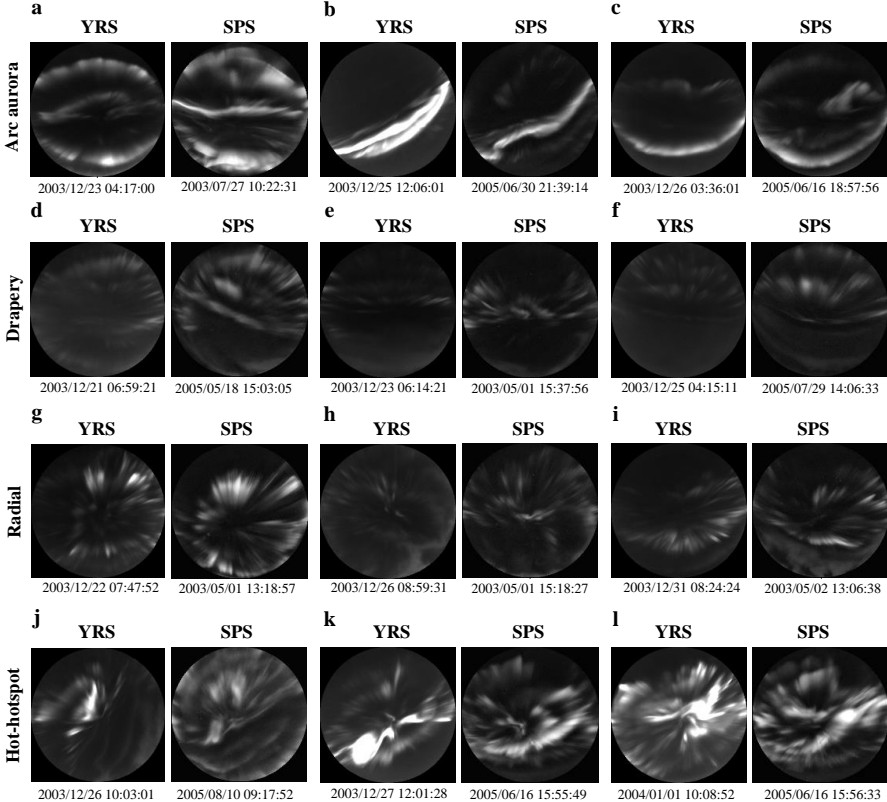

**Figure 3. Comparison of the four auroral types at SPS and YRS by retrieval experiments. The images from labeled ARD dataset of YRS are used as query images, and the images of SPS are the retrieval results from ATD1 dataset which are the most similar with each query image.**

**3.2.2 Supervised classification**

Based on the spectral signatures of dayside auroras, Hu et al.(2009) partitioned the dayside oval into four auroral active regions, including prenoon warm spot ("W", 0730-0930MLT), the midday gap ("M", 0930-1300MLT), the postnoon hot spot ("H", 1300-1530MLT), and the dusk aurora sector ("D", 1530-1700MLT). In this section, supervised classification experiments are conducted on ARD dataset to estimate the classification accuracy in different auroral active regions.

Specifically, the widely used $k$-nearest neighbor ($k$-NN) classifier is selected. A dataset is divided into two subsets: the training set (suppose image type is known) and testing set (suppose image type is unknown). An image in the testing set is classified by a majority vote of its neighbors in the training set and assigned the most common class among its $k$-nearest neighbors ($k$ is a positive integer). The value choice of $k$ is very critical for $k$-NN classifier, the smaller the $k$, the result is more sensitive to the noise and the overlap between classes, causing unstable results; the greater the $k$, the majority class of the dataset will dominate the neighbors, resulting classification error. Considering the size of ARD dataset is 8001, we conducted experiments using $k$=1, 3, and 25, respectively.

10-fold cross-validation experiments are performed on ARD dataset. Images of the four auroral

types are separately divided into 10 parts randomly, of which 9 parts are used to train the classifier, and
the remaining 1 part are used as the testing set to evaluate the classification effectiveness. The
training-testing ratios in each auroral active region are not strictly guaranteed to be 9:1. Table 1 shows
their specific number. In order to obtain robust results, the experiments are repeated 100 times with
different data partition manners. The mean accuracy and standard deviation are given in Table 2.
**Table 1.**   Image number of the four auroral types in different auroral active regions, including the
number of the testing set of 10-fold cross-validation experiment and the number of all ARD dataset (in
the bracket).

| type \\ regions | Arc | Drapery | Radial | "hot-spot" |
|---|---|---|---|---|
| "W" | 71(726) | 38(413) | 16(166) | 4(33) |
| "M" | 30(282) | 118(1108) | 100(1037) | 27(259) |
| "H" | 105(992) | 11(136) | 28(254) | 42(442) |
| "D" | 102(1101) | 4(24) | 0(0) | 4(42) |

**Table 2.**   100 times experiment performance (mean accuracy±standard deviation%) of $k$-NN classifier
in different auroral active regions.

| classifier | type \\ regions | Arc | Drapery | Radial | "hot-spot" |
|---|---|---|---|---|---|
| NN | "W" | 98.37±1.73 | 95.96±2.99 | 93.29±5.82 | 91.74±16.94 |
|  | "M" | 94.17±4.21 | 96.83±1.50 | 95.99±1.88 | 93.15±5.75 |
|  | "H" | 98.83±1.04 | 95.66±5.79 | 90.52±5.79 | 94.89±2.99 |
|  | "D" | 99.27±0.71 | 90.82±19.83 | 96.00±19.69 | 76.34±20.69 |
| 3-NN | "W" | 98.83±1.29 | 94.42±3.87 | 88.36±7.38 | 84.23±20.30 |
|  | "M" | 91.74±4.99 | 94.61±2.05 | 92.53±2.47 | 87.88±6.47 |
|  | "H" | 98.00±1.28 | 91.83±8.22 | 85.42±6.81 | 94.46±3.46 |
|  | "D" | 98.92±1.08 | 83.08±23.97 | 78.00±41.63 | 80.31±23.54 |
| 5-NN | "W" | 98.37±1.36 | 92.78±4.22 | 86.66±8.29 | 81.89±23.53 |
|  | "M" | 91.35±5.47 | 92.69±2.38 | 90.75±2.79 | 86.68±7.23 |
|  | "H" | 97.44±1.31 | 88.62±10.29 | 84.22±7.06 | 92.68±4.15 |
|  | "D" | 98.19±1.40 | 79.80±28.87 | 77.00±42.30 | 78.07±25.34 |

From Table 2, we can conclude that (1) NN classifier works best. Many two adjacent images in
the ARD dataset are picked out in a short time interval. Specifically, there are 1690 out of 8001 images
captured with the intervals less than 2 minutes (Yang et al., 2012). ASC images with such short
intervals always have similar morphology, therefore the NN classifier works best in the experiments. (2)
In the "W" region, the classification accuracy changes a lot and has a big standard deviation value. This
is because there are very few "hot-spot" auroras (less than 4% of the total number), as shown in Table 1,
even only one image is categorized into different types, the accuracy will change a lot. (3) In the "D"
region, the classification accuracy also changes a lot and has a big standard deviation value too. The



reason is also presented in Table 1, most images in the "D" region are arc auroras, and very few images
belong to other auroral types (less than 8% of arc auroras). Very few images being classified into
different categories may cause the accuracy of drapery corona, radial corona and "hot-spot" auroras
changes a lot and with big deviation values. (4) On the whole, the more the image data, the smaller is
the deviation value. (5) Except those invalid results (standard deviation is greater than 10%, which is
labeled in gray shadow in Table 2), the accuracy of arc and "hot-spot" auroras has a significantly
decrease at the region of "M", while drapery and radial coronas have a lower accuracy at the region of
"H". (6) The proposed method achieves very good performance: almost all the classification accuracies
are higher than 90%.
**3.2.3 Occurrence distributions**
In this part, the labeled ARD dataset is performed as the training set, and by exploiting the class
information contained in it, we recognize all the images in ATD1 and ATD2 by a $k$-NN classifier. The
occurrence distributions of the four auroral types classified by $k$-NN are plotted and compared at both
stations.
Unlike ARD, when constructing ATD1 and ATD2, all auroral images are picked except for the
images captured under bad-weather conditions or having no aurora information, therefore the
classification rejection is needed. The $k$-NN method classifies an object by a majority vote from its
neighbors, if there is no majority agreement, the testing image is rejected to be assigned a label. In the
occurrence distribution experiments, if an image in ATD1 or ATD2 is discarded by $k$-NN classifier, it
is labeled as unknown. In consideration of the size of ATD1 and ATD2, we consider $3$-NN and $25$-NN
classifiers, respectively. The detailed results are given in Table 3.
**Table 3.** Classification results on ATD1 of SPS and ATD2 of YRS using 3-NN and 25-NN
classifiers.

| classifier | station (dataset) | type number/ ratio | Arc | Drapery | Radial | "hot-spot" | unknown |
|---|---|---|---|---|---|---|---|
| 3-NN | SPS (ATD1) | number | 15824 | 13110 | 7661 | 1124 | 1616 |
| | | ratio | 0.402 | 0.333 | 0.195 | 0.029 | 0.041 |
| | YRS (ATD2) | number | 24913 | 23027 | 12920 | 2572 | 1929 |
| | | ratio | 0.381 | 0.352 | 0.198 | 0.039 | 0.030 |
| 25-NN | SPS (ATD1) | number | 15447 | 14956 | 7471 | 628 | 833 |
| | | ratio | 0.393 | 0.380 | 0.190 | 0.016 | 0.021 |
| | YRS (ATD2) | number | 25473 | 24231 | 12827 | 1677 | 1153 |
| | | ratio | 0.390 | 0.371 | 0.196 | 0.026 | 0.018 |

From Table 3 we can see, (1) there is little difference between 3-NN and 25-NN classifiers,
especially for radial corona, arc, and hot-spot auroras, the difference of which is less than 1.5%; (2) the
most different auroral type between the two stations is drapery corona, because the texture of the
drapery aurora is complex and error-prone. (3) The occurrence percentages of the four auroral types at
both stations in northern and southern hemispheres are very close, of which the arc aurora is about 39%,



while drapery corona is around 33%-38%, radial corona is about 20%, and hot-spot is around 3%. (4)
The number of the "unknown" type obtained by the 25-NN classifier is less than the 3-NN, because it is
more difficult for 3-NN classifiers to get a majority vote.
The temporal distributions of the four auroral categories are presented in Figure 4. The temporal
axis is divided into 39 bins (/36 bins) with 20-minute durations for ATD1 (/ATD2) for better display.
Image number within each bin for each category is first counted, and by normalizing the total image
number in the same bin, the occurrence distributions are obtained. There are no significant differences
between the distributions obtained by 3-NN and 25-NN, so we only show that of the 25-NN in Figure 4.
The first panel of Figure 4 shows the distribution of all images in ATD1 (Figure 4(a)) and ATD2
(Figure 4(b)). The bins around 12 MLT have the minimum image numbers at both SPS and YRS,
because optical observations near noon are tend to be disturbed by sunlight.
From the second to bottom panel, Figure 4 shows the occurrence distribution of the four auroral
categories and the unknown type respectively. At the very top of Figure 4, four active regions proposed
by Hu et al. (2009) are distinguished by bold dashed lines, while the two states of the midday gap are
partitioned by a thin dashed line. The four auroral types dominate different dayside oval regions with
the peaks fall into each region respectively.
From Figure 4, we can see the occurrence distributions at the both stations are similar: (1) The
occurrence distributions of arc auroras show a distinct asymmetric double-peak between pre-noon and
post-noon (Akasofu and Kan, 1980; Liou et al., 1997; Meng and Lundin, 1986; Newell et al., 1996; Hu
et al., 2009). (2) Two corona auroras, drapery and radial, have similar auroral morphologies, and both
dominantly occur before 1300MLT. (3) The fast changing hot-spot auroras most occur in region H and
have a distinct small peak around 1330MLT. (4) There are a few atypical images in ATD1 and ATD2
that are classified as unknown. (5) In addition to the similar distribution trends of the four auroral
categories at both stations, the occurrence ratios of each category in each MLT at both stations are also
very close.
Although there are so many similarities, differences also exist in the auroral occurrence
distributions of the two stations: (1) Drapery and radial coronas show different peak positions at the
two stations. (2) Although the "hot-spot" aurora has an occurrence peak around 1330MLT at both SPS
and YRS, the duration of YRS is much longer than SPS.



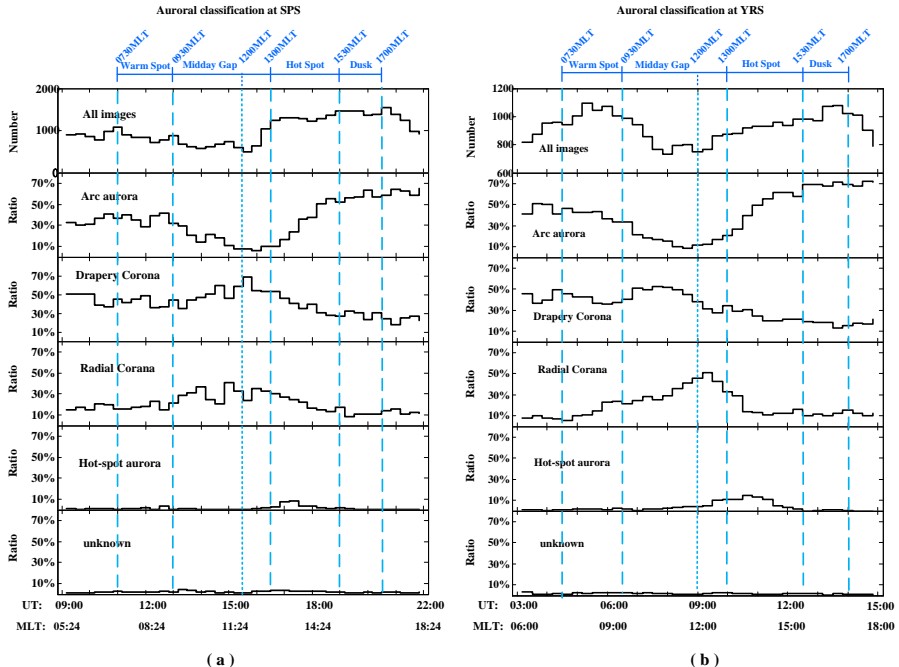

**Figure 4. Temporal occurrence distributions of auroral types at SPS (Figure 4(a)) and YRS (Figure 4(b)): from top to bottom panels show the distributions of image numbers (top panel), four categories of dayside auroras (2nd-5th panels) in ATD1 and ATD2, and the unknown images in both dataset, respectively.**

**4 Discussion**

The discrete aurora includes acceleration mainly caused by two physical mechanisms, one is quasi-static electric fields, producing monoenergetic auroral precipitation that always appears in the "inverted V" electron spectrum; the other is dispersive Alfvén waves, producing broadband auroral electron precipitation (Newell et al., 2009). Ionospheric satellite observations show that monoenergetic precipitation most exists in the 14-19 MLT region of post-noon auroral oval, followed by the 06-09 MLT region of pre-noon auroral oval (the incidence and electron precipitation energy flux of the latter is lower than the former), and the noon 09-14MLT is the least likely region for observing mono-energetic electron acceleration (Newell et al., 1996, 2009). Such a distribution is consistent with that of the arc aurora. As shown in Figure 4, at both stations, the occurrence percentages of arc aurora at 14-18MLT region of post-noon are greater than 50%, and the percentages at 06-09MLT region of pre-noon are 30%-50% while less than 30% at the noon of 09-14MLT, especially around 12MLT of the noon the percentages are even less than 10%. The similar occurrence distributions between the two stations demonstrate that the arc auroras with electron spectrum characteristics of "inverted V" structures are closely associated with quasi-static electric fields acceleration.

Dayside coronas (include drapery, radial and "hot-spot") have distinct filament structures, which indicates that (1) the corona aurora is excited in a broad altitude range, and (2) the energy distribution





of these precipitating electrons that excite coronas are very broad (The excitation altitude of aurora is
closely related to the energy of precipitating electrons: the higher the energy, the lower the altitude of
precipitating electrons enter into ionosphere). Satellite spectrum probe has proved that the precipitating
electron spectrum of dayside coronas has the signatures of the broadband auroral electron precipitation
(Hu et al., 2009). Ionospheric satellite observations show that the broadband electron precipitation in
the dayside auroral oval most occurs at the 06-15MLT region, and occurs more often at pre-noon than
post-noon (Newell et al., 2009). Occurrence rate of dayside corona auroras at both stations in northern
and southern hemispheres also dominates 06-15MLT regions, which is similar with that of the satellite
detection. In addition, satellite observations obtained at a higher altitudes (>6000km) show that
dispersive Alfvén waves primarily occur at the 06-15MLT region of the dayside auroral oval (Chaston
et al., 2007). Therefore, dayside corona auroras are closely related to dispersive Alfvén waves.

**5 Conclusion**

Based on the previous studies of morphological classification to dayside auroras (Hu et al., 2009),
and by applying image processing and pattern recognition techniques on the ASC observations at SPS
(during years 2004-2006) and YRS (during years 2003-2009), this paper made an automatic
recognition of dayside auroral morphology in southern and northern hemispheres and a statistic
analysis to the distribution of dayside auroral types. Experimental results show that in both southern
and northern hemispheres, the dayside arc auroras primarily occur at post-noon (14-18MLT) and
pre-noon (06-09MLT) regions and most occur at post-noon, while between the two peaks, the noon
region (09-14MLT) forms a "midday gap". Dayside corona auroras most occur at 06-15MLT regions.
The distribution of arc and corona auroral types corresponds to the occurrence rate of quasi-static
electric fields acceleration and dispersive Alfvén wave acceleration on dayside auroral oval
respectively. However, the ground-based optical observations demonstrate that the dayside corona
auroras can be classified into three types, including drapery corona auroras, radial corona auroras, and
"hot-spot" auroras. These corona auroral types are possibly related with the propagation process of
dispersive Alfvén wave at different magnetosphere boundary layers, and results in the difference of the
three corona auroral types between the two stations. Such an inference needs further confirmation by
combination analysis of satellite and ground-based optical observations.

**Acknowledgements**

We acknowledge Dr. Yusuke Ebihara at Kyoto University to provide the ASC auroral observation
data of the South Pole Station (http://www.southpole-aurora.org/). This work was supported by the
National Natural Science Foundation of China (41504122, 41274164, 41431072, 41374159), Shaanxi
Province youth talent fund (2016JQ4001), Young Talent fund of University Association for Science
and Technology in Shaanxi, China (20160211), the Polar Environment Comprehensive Investigation
and Assessment Programs (CHINARE2017-02-04, CHINARE2017-02-04), the Strategic Priority
Research Program on Space Science, the Chinese Academy of Sciences (XDA15350302). Data issued
by the Data-sharing Platform of Polar Science (http://www.chinare.org.cn) maintained by Polar
Research Institute of China (PRIC) and Chinese National Arctic & Antarctic Data Center (CN-NADC).



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
