# Peer review of "A comparative study of auroral morphology distribution between northern and southern hemispheres based on automatic classification"

_Geoscientific Instrumentation, Methods and Data Systems, 2017_

## Referee Comment (RC1) · Anonymous Referee #2 · 28 Nov 2017

Review of GI-2017-49

Yang and Hu, "A comparative study of auroral morphology distribution between northern and southern hemispheres based on automatic classification"

–

The authors describe experiments with data from two auroral cameras with one being in the northern hemisphere and the other in the southern hemisphere. The central idea is to use a KNN classifier to compare the auroral type and to construct occurrence distributions from the data. The topic is very relevant to the GI journal and a welcome contribution to data analysis methods in space research.

In general, I feel that the manuscript would benefit from a language check by a native English speaker to improve the "flow". There are a few spots where interpreting what the authors want to say becomes difficult.

My main critisism is that the classification results are "too good". Earlier studies, including the listed references (Wang et al., 2010, Syrjasuo and Donovan, 2004) have highlighted the uncertainty of auroral categories. The issue is whether a fixed number of auroral classes such as "arc", "drapery", "radial" and "hot-spot" is sufficient for all auroral shapes one encounters in the data. A simple method like the KNN will always select K neighbours and thus always provide a label regardless of input data. From the results (table 2) I do get a feeling that the four auroral types can be separated quite well. But what happens if one tries to classify an image that does not belong to any of these classes? An obvious example would be diffuse aurora, which does meet the criteria of not being taken during bad weather and having aurora in the sky (although not necessarily daytime...). This is very important from the practical point of view: if we can rely on the classification, we can also consider, for example, higher time resolution, if something "interesting" happens in the sky.

So, I recommend a major revision with more depth in the analysis of the retrieval and classification itself.

1) Image retrieval (Fig 3): rather than showing only the closest match, which always exists, it would be much better to show, for example, five or more closest matches with their Chi-squared distances. What are your observations, are all images "matches"? At what distance do you think the images are not any more similar?

For this analysis to work, you most likely need to consider the capture time of each retrieved image to separate the similarity due to capture time rather.

2) Machine learning algorithms such as KNN are known to be rather bad in extrapolating and then there is the infamous "curse of dimensionality" (sampling density). So, my main question is whether one can always rely on the classification result. Answering this does require some additional work. KNN is a special case of kernel density estimation and there are thus built-in uncertainties which should be examined.

a) Your dataset ARD contains four types of aurora. What is the variability/distribution of pair-wise distances within one class? I would expect arcs to be a "tight group" but all of the other types may be more scattered. What is your observation and conclusion about how it may affect your results?

b) What is the variability/distribution between the distances between different types of aurora? Are the types of aurora well separated? If not, should that be taken into account in KNN by, e.g., ignoring far away neighbours? While requiring all neighbours to be from the same class does improve the reliability, it still chooses the closest class (whether it belongs there or not). For example, I do not see any obvious difference between the "radial" (Fig 3g) and "hot-spot" (Fig 3j) types of aurora.

————-

3) How does the viewing geometry affect your classification? And does it matter for this study?

===========

Minor comments:

4) Abstract, line 9: the authors were perhaps thinking of "important" rather than "significant"?

5) Introduction, line 30: please clarify the sentence containing "the inversion of magnetic layer structure and dynamic process"

6) Introduction, line 33: replace "our country" with, for example, "Chinese research institutes".

7) Introduction, line 52: what do you mean with "multiple-wavelength intensity distributions"? Is it relevant for this manuscript?

8) Data and methodology, line 76: I am unsure what exactly you mean with "MLT~UT-3.6 hrs."

9) SPS data and YRS data: in my opinion, the description of preprocessing steps and the instruments would benefit from a table which would list the technical details for each instrument. As far as I see the process starts with first manually removing data without aurora and then rotating and scaling the images for classification. For YRS, there is also substraction of dark current (which is assumed to be the same for the whole image?). So, the steps are almost identical for both cameras and just the rotation angles etc. are different.

10) SPS data, line 84: your statement "cannot work normally in cloudy or foggy weather" is incorrect as the instruments work fine in those conditions – you just cannot see the aurora! If I interpret the manuscript correctly, you (manually) select only those images that contain aurora. Please clarify.

11) SPS data, line 89; YRS data, lines 564 and 117: what are the dynamic ranges of these cameras? Providing a number "4000" does not mean anything in itself. My understanding of intensity stretching uses a minimum pixel value (black point) and the maximum pixel value (white point). Is this the maximum and you start from zero (black)? Please clarify.

12) SPS and YRS data: why are you carrying out the image stretching? LBP uses relative intensity values and should thus be insensitive to brightness changes as long as the order of brightness is maintained. Please clarify.

13) ASC image representation, lines 119-135: Wang et al. (2010) used an improved LBP and carried out experiments to determine best parameters for LBP neighbourhood size (and number of samples). They concluded that the parameter choice was rather insensitive but they are essentially using a 5x5 neighbourhood, from which 8 pixels are sampled at a radius of 2 pixels from the centre pixel. On the other hand, you are using 3x3 neighbourhood. Could you add some text to explain why you chose to do it your

way and what the advantages/disadvantages are, please?

14) Classification mechanism, line 156: how well does this 1-min time interval work with arcs that may be present for a longer time (also in the daytime)?

15) Supervised classification: how is this different from what Wang et al. (2010) did? Are the labels now "W", "M", "H" and "D"? What about those images that have aurora whose type is not arc, drapery, radial or "hot-spot"?

16) Supervised classification, line 192: Please clarify your choice k=1,3,25. Why 25 rather than sampling closer to 3?

17) Table 1. Please revise the caption of table ("image number" means the number of images?)

18) Table 2. There is a bias in all error analysis methods and crossvalidation is no expection: please reconsider the precision of classification accuracies (is 98.37% really meaningful or should it be 98% due to errors in estimating the error?)

19) Occurrence distribution, line 250: you indicate (lines 81 and 103) that both cameras can operate 24-h in the wintertime. Yet this is clearly not fully correct as there are fewer images around the noon due to sunlight. Please be more specific and revise the text accordingly.

20) Occurrence distribution, line 260: is this description of "fast changing" hot-spot auroras coming from the data? You were analysing single images without temporal analysis, weren't you?

21) Conclusions, lines 313-314: your sentence "all dayside coronal auroras can be classified into three auroral types" is a very strong statement. Given my concerns about the number of possible (correct) auroral types, I don't quite agree yet. I leave it to the the authors to consider whether this statement belongs to "discussion" rather than conclusions.

————

An observation, not a comment on the manuscript:

The authors might want to consider using their method to study the lifetimes of dayside auroral forms in a future study: it would be interesting to use Chi-squared distance to locate time periods when the forms are stable (or changing rapidly). A simple method would be to create a time-series of the distance to the previous image to study the evolution of similarity in time.

---

## Referee Comment (RC2) · Anonymous Referee #1 · 4 Dec 2017

Looks this paper is enough good for publishing.

---

## Author Comment (AC1) · 5 Jan 2018

Based on the previous observations and investigation, we summarized the dayside discrete auroras into four types, i.e., the dayside arc, dayside radial corona, dayside drapery corona, and dayside hot-spot aurora.

Using the auroral images of YRS, we manually picked out typical auroral images (namely the ARD dataset) of the four types dayside discrete aurora, which consists of 8001 auroral images from December 2003 to January 2004. Based on the ARD dataset, we do the supervised experiments. The training-testing ratio is very high, i.e., 9:1, and got the typical characteristics of the four types of dayside auroras.

[Figure]

In this paper, we just study the dayside discrete aurora, the diffuse aurora in dayside oval is not belong to the four types of dayside discrete aurora. Therefore, we added an "unknown" type for the diffuse aurora and other auroral forms which we could not classify.

We are very grateful to the comment of Anonymous Referee #2. We'll modify the manuscript based on the comments. In addition, the answers for major and minor comments are as the follows:

The answers for the major comments:

1) Image retrieval (Fig 3): rather than showing only the closest match, which always exists, it would be much better to show, for example, five or more closest matches with their Chi-squared distances. What are your observations, are all images "matches"? At what distance do you think the images are not any more similar? For this analysis to work, you most likely need to consider the capture time of each retrieved image to separate the similarity due to capture time rather.

Answer: We have made the five or more closest matches (rank1, rank3, rank5, rank7 and rank 9) with their Chi-squared distances. Some examples are shown in the fig. 1 and fig. 2. Fig. 1 and fig.2 are the table of Chi-squared distances, and aurora images corresponding to the content of the table, respectively. Two query images are given for each auroral class (fig 02). From these examples, we can find that the distance is smaller when the image is more similar with the query image. However, it is hard to set a threshold that once exceed the distance the images are not any more similar. Because the distance of un-similar images is different for different auroral categories, and different auroral images.

2) Machine learning algorithms such as KNN are known to be rather bad in extrapolating and then there is the infamous "curse of dimensionality" (sampling density). So, my main question is whether one can always rely on the classification result. Answering this does require some additional work. KNN is a special case of kernel density

estimation and there are thus built-in uncertainties which should be examined. a) Your dataset ARD contains four types of aurora. What is the variability/distribution of pair-wise distances within one class? I would expect arcs to be a "tight group" but all of the other types may be more scattered. What is your observation and conclusion about how it may affect your results? b) What is the variability/distribution between the distances between different types of aurora? Are the types of aurora well separated? If not, should that be taken into account in KNN by, e.g., ignoring far away neighbours? While requiring all neighbours to be from the same class does improve the reliability, it still chooses the closest class (whether it belongs there or not). Answer: (1) Traditional classification includes pre-processing, feature extraction and classifier design, of which the feature extraction is the most critical step. Therefore, we just consider the simple and widely used kNN classifier. (2) The variability/distributions of pair-wise distances within one class and between different types are shown below (fig 03). This figure is the pseudo color image of Chi-squared distance matrix. Number 1$\sim$3933 are arc aurora, Number 3934$\sim$5719 are drapery aurora, Number 5720$\sim$7219 are radial, and Number 7218$\sim$8001 are hot-spot aurora. The color represents the value of distance. From this figure, we can find there is a "tight group" in drapery and radial corona auroras but arcs are more scattered. Arc auroras have good classification performance but present an unexpected "scattered group". We think possibly it means that the arc images form a continuum, where it is always possible to find images with features in between the ones we use for our classification; on the other hand, the very different shape and texture indicate that arc auroras can be further divided into several subclasses. (3) Yes, k should not be a very large value. In this manuscript, the value of k is considered according to the capture time of the observations in the dataset. ARD dataset are constructed by extending the interval between adjacent images to $\sim$1minute, so we consider k=1,3,5; and for ATD1 and ATD2 dataset, the time interval is $\sim$10 seconds, therefore k=1,3,25 are considered.

3) How does the viewing geometry affect your classification? And does it matter for this study? Answer: Auroral images captured at South Pole station and Yellow River station

can be seen as from different viewing geometry. We don't think it has many influences on this study. However, the sensitivity of ASC can affect the study, because some textural features of the dayside auroras could not be captured by the low-sensitivity ASC.

The answers for parts of minor comments are as the follows. Other minor comments are answered in the new revised manuscript:

11) SPS data, line 89; YRS data, lines 564 and 117: what are the dynamic ranges of these cameras? Providing a number "8000" does not mean anything in itself. My understanding of intensity stretching uses a minimum pixel value (black point) and the maximum pixel value (white point). Is this the maximum and you start from zero (black)? Please clarify. Answer: Yes, 8000 is set by using the trial-and-error method, and we just want to make the auroral images visually clear. After stretched with a cutoff value of 8000, the pixel values are starting from zero to 8000. 12) SPS and YRS data: why are you carrying out the image stretching? LBP uses relative intensity values and should thus be insensitive to brightness changes as long as the order of brightness is maintained. Please clarify. Answer: For the feature extraction point of view, the image stretching is unnecessary. The purpose of carrying out the image stretching is just easy for human visual judgment. And it does not influence the other results.

13) ASC image representation, lines 119-135: Wang et al. (2010) used an improved LBP and carried out experiments to determine best parameters for LBP neighborhood size (and number of samples). They concluded that the parameter choice was rather insensitive but they are essentially using a 5x5 neighborhood, from which 8 pixels are sampled at a radius of 2 pixels from the center pixel. On the other hand, you are using 3x3 neighborhood. Could you add some text to explain why you chose to do it your way and what the advantages/disadvantages are, please? Answer: 3x3 neighborhood is the basic LBP, and 5x5 neighborhood is the improved LBP. For easy explaining the idea of LBP, Figure1 just shows the calculation process of the basic LBP operator. In this paper, we also used the improved LBP, which is the same as Wang et al. (2010).

We add this in the revised manuscript.

14) Classification mechanism, line 156: how well does this 1-min time interval work with arcs that may be present for a longer time (also in the daytime)? Answer: We just classify the discrete dayside aurora based on the single image (static characteristics of dayside discreate aurora), not sequence images (dynamic characteristics of dayside discrete aurora). In 1-min interval, the characteristics of dayside discrete aurora have obvious changes. Therefore, we can get more static characteristics of dayside discrete aurora when we use short interval images, i.e. 1-min interval image. The living time of arc is often longer than 1 minute, the 1-min interval can get more static characteristics of the arcs.

15) Supervised classification: how is this different from what Wang et al. (2010) did? Are the labels now "W", "M", "H" and "D"? What about those images that have aurora whose type is not arc, drapery, radial or "hot-spot"? Answer: In the supervised classification, the methods used are the similar with Wang et al. (2010), the difference is: we use the standard cross-validation technique to assess the classification method. Specifically, 10-fold cross-validation is used, and the training-testing ratio is different from what Wang et al. (2010) set. The labels of auroral images are arc, drapery, radial and hot-spot. We partition the dayside oval into four auroral active regions ("W", "M", "H" and "D") according to their capture time for further analysis. Those images that have aurora whose type is not arc, drapery, radial or "hot-spot" are refused to classify by k-NN classifier, and we call these "unknown" type in the manuscript.

18) Table 2. There is a bias in all error analysis methods and cross validation is no expection: please reconsider the precision of classification accuracies (is 98.37% really meaningful or should it be 98% due to errors in estimating the error?) Answer: 98.37% is the mean value. The bias is the standard deviation, i.e., root mean squared error.

[Figure]

| Query | Rank 1 | Rank 3 | Rank 5 | Rank 7 | Rank 9 |
|---|---|---|---|---|---|
| Arc Aurora(1) | 29290.9 | 30112.3 | 31104.1 | 31542.5 | 32085.8 |
| Arc Aurora(2) | 22083.6 | 23117.6 | 23806.9 | 24051.1 | 24359.6 |
| Drapery Aurora(1) | 19388.6 | 20359.7 | 20681.8 | 21074.8 | 21110.4 |
| Drapery Aurora(2) | 16812.3 | 17020.9 | 17106.7 | 17345 | 17395.5 |
| Radial Aurora(1) | 25498.8 | 27769.7 | 27859.3 | 27954.8 | 28467 |
| Radial Aurora(2) | 20773.5 | 21848.2 | 22099.8 | 22117.7 | 22432.3 |
| Hot-spot Aurora(1) | 27485.9 | 28564 | 28862.8 | 29029.9 | 29308.9 |
| Hot-spot Aurora(2) | 28582.8 | 33475.6 | 33537.2 | 33864.2 | 34040.4 |

**Fig. 1.** Figure 01

[Figure]

**Fig. 2.** Figure 02

[Figure]

**Fig. 3.** Figure 03